# Using numerical modeling and simulation to assess the ethical burden in clinical trials and how it relates to the proportion of responders in a trial sample

Jean-Pierre Boissel[1]☉*, David Pérol[2]☉, Hervé Décousus[3]☉, Ingrid Klingmann[4]☉, Marc Hommel[1,5]☉

1 Novadiscovery, Lyon, France, 2 Department of Biostatistics, Centre Léon Bérard, Lyon, France, 3 INSERM, CIC 1408—F Crin, INNOVTE, CHU Saint-Etienne, Hôpital Nord, Service Médecine Vasculaire et Thérapeutique, Saint Etienne, France, 4 European Forum for Good Clinical Practice (EFGCP), Brussels, Belgium, 5 University Hospital Grenoble, Grenoble, EA 4407 AGEIS UGA, France

☉ These authors contributed equally to this work.

* jean-pierre.boissel@novadiscovery.com

**Data Availability Statement:** Data resulting from simulations are available as supplementary material.

## Abstract

In order to propose a more precise definition and explore how to reduce ethical losses in randomized controlled clinical trials (RCTs), we set out to identify trial participants who do not contribute to demonstrating that the treatment in the experimental arm is superior to that in the control arm. RCTs emerged mid-last century as the gold standard for assessing efficacy, becoming the cornerstone the value of new therapies, yet their ethical grounds are a matter of debate. We introduce the concept of unnecessary participants in RCTs, the sum of non-informative participants and non-responders. The non-informative participants are considered not informative with respect to the efficacy measured in the trial in contrast to responders who carry all the information required to conclude on the treatment's efficacy. The non-responders present the event whether or not they are treated with the experimental treatment. The unnecessary participants carry the burden of having to participate in a clinical trial without benefiting from it, which might include experiencing side effects. Thus, these unnecessary participants carry the ethical loss that is inherent to the RCT methodology. On the contrary, responders to the experimental treatment bear its entire efficacy in the RCT. Starting from the proportions observed in a real placebo-controlled trial from the literature, we carried out simulations of RCTs progressively increasing the proportion of responders up to 100%. We show that the number of unnecessary participants decreases steadily until the RCT's ethical loss reaches a minimum. In parallel, the trial sample size decreases (presumably its cost as well), although the trial's statistical power increases as shown by the increase of the chi-square comparing the event rates between the two arms. Thus, we expect that increasing the proportion of responders in RCTs would contribute to making them more ethically acceptable, with less false negative outcomes.

**Funding:** This work was supported by the RESSTORE project funded by the European Commission under the H2020 program (Grant Number 681044). The funders had no role in study design, data collection and analysis, decision to publish, or preparation of the manuscript.

**Competing interests:** All authors have completed the Unified Competing Interest form (available on request from the corresponding author) and declare: no support from any organization for the submitted work [or describe if any]; no financial relationships with any organizations that might have an interest in the submitted work in the previous three years [or describe if any], no other relationships or activities that could appear to have influenced the submitted work [or describe if any]. This does not alter our adherence to PLOS ONE policies on sharing data and materials.

## Introduction

Ethical loss in randomized controlled clinical trials (RCTs) is an intuitive and rather vague concept, which can be outlined at a first glance as a loss of chance, i.e. everything an individual either misses or experience as detrimental as a result of participating in a RCT. The uncertainty associated with trial enrollment has two opposite issues. In an attempt to address these two opposite uncertainties, Freedman proposed the concept of "equipoise" [1]. RCTs remain the gold standard for evaluating the efficacy and safety of new therapies [2,3]. Despite their wide acceptance and applying the principle of 'equipoise', some questions of ethics in RCTs remain unanswered. There are various types of ethical concerns linked to randomization: prior to launching a trial of a new treatment, randomizing participants to conventional treatments that is deemed less effective is a breach of the equipoise principle, just as happen with allocation to a placebo, or to a known sham treatment; once the trial is completed, if the experimental treatment turns out to be either more effective or less effective (or even potentially harmful), participants in the control arm or those in the experimental arm, respectively, face a loss of chance. If the required informed consent is usually presented as a measure to overcome [4] these issues, in reality it potentially transfers part of these concerns from investigators to participants. Several alternatives to traditional RCTs have been proposed recently to address some of these ethical concerns: adaptive designs enabling planned interim analyses that can lead to a reduction of the total number of randomized participants or to a deletion of study arms with inefficient treatment regimens and re-randomization of subjects to the more promising treatment arms; and external control arms that suppressed the need to randomize participants to the 'conventional treatment' [5,6].

It is known that when a new treatment is being trialed, some participants in the treatment arm will present the event the treatment is supposed to prevent, as opposed to the "responders", i.e., those showing the expected reaction to treatment. Identification of responders is a timely and challenging issue that remains poorly studied, mainly because it is difficult to predict how participants will respond to the experimental treatment. But beyond the importance of identifying responders in the era of personalized medicine, the consequences of enrolling non-responders in RCTs remain unexplored.

Although intuitively, selecting responders in RCTs is a worthwhile endeavor, the benefit of designing trials that meet this goal remains unexplored. Further, there is a lack of consensus in recommending it. Large trials, recruiting with little constraints have been advocated and frequently undertaken [7,8]. Regulators advise that criteria over inclusion and exclusion criteria in phase III trials should be relaxed as much as possible [9], while suggesting maintaining sufficient homogeneity, seeking for a compromise that is not easy to achieve.

The work presented here focus on the ethical issue arising from including non-responders in RCTs. More precisely, the objective of our work is to show how RCT simulations can be used to address the consequences of diluting the number of responders in the trial sample. Our simulations do not aim at quantitatively defining adequate parameters of an RCT but intend to draw upon the qualitative perception of the issue of ethical loss linked to the current way of how RCTs are planned, especially when it comes to eligibility criteria. To that extent, our approach focuses on treatment efficacy. We postulate that identifying responders could reduce uncertainty and could be viewed as an operational equipoise principle.

## Material and methods

### Overview

Collected data from a published trial was used for initiating the simulations conducted for the purpose of this study. The definitions of each category of trial participants defined below were

applied to allow for computing the required number of participants for the five categories. The changes in different trial characteristics as a function of the proportion of responders was explored, taking the published trial as the point zero.

## Conceptual framework of this work

**The two separate objectives of a clinical trial.** When one states that a trial is "aimed at demonstrating the efficacy of X in, for example, preventing death in patients with type 2 diabetes", the statement has two intertwined meanings, or objectives, one being qualitative, the other quantitative. First, it means that the trial is expected to show that X is more effective than the control in preventing death (qualitative objective), and second, that it will provide an estimate of the effect size (quantitative objective). The two objectives are tightly intertwined because the achievement of the first one depends, among others, on the observed effect size. However, for the sake of our reasoning, these must be strictly separated, focusing on the number of responders (a quantitative objective). Although out of the scope of the work developed here, there are other, more fundamental, reasons for separating these two objectives. Just to name two examples: while the efficacy is a treatment property, which supports consideration of extrapolating the context of use, the validity of the effect size estimate based on the difference between the observed rates of the event in the two arms is limited to the trial itself.

## "Responders", "non-responders", "non-informative", and "unnecessary" participants

**Definition of a responder.** Let us assume that the expected effect of the treatment of interest is to prevent a dichotomous (yes/no) event, such as death. In this instance, a responder is a patient who would have presented the event the experimental therapy is supposed to prevent if he/she had not been treated with this therapy, and who will not present it if treated with the therapy. In parallel-arm design, RCT with mortality endpoints, responders are subjects with a fatal outcome under control treatment (C) administration but who are kept alive if administered the experimental treatment (T). The control treatment can be the standard of care (SoC) alone, a competitor or a placebo, both on top of SoE. Although it sounds simple, in practice, this definition of responders is to today difficult, if not impossible, to apply: how to predict that a given patient is a responder? Note the assumption that the experimental treatment is better than the control, at least for a few participants, or even a single one.

*Number of responders in a two-arm RCT.* Let us use as an example a two-arm completed RCT with the number of event-free subjects and participants having presented the event as shown in Table 1. At baseline, the two study arms are supposed to be identical, thanks to randomization. Although in the real world, the blind randomizing process is aimed at producing two groups identical on average, we will assume that all subjects included in the trial display similar baseline characteristics, i.e. are exchangeable.

The event rates are defined as $R_c = a/N_c$ and $R_T = b/N_T$ in the control and experimental treatment groups, respectively. The experimental treatment efficacy is measured by the

**Table 1. Summary data of a completed RCT.** *a*, *b*, *c*, *d* are number of patients in each cell and Nc and NT the total number of patients in the control and the experimental arms respectively.

|  | Control arm (C) | Experimental treatment arm (T) |
|---|---|---|
| **Event** | *a* | *b* |
| **No event** | *c* | *d* |
| **Total** | Nc | NT |

absolute benefit, AB = Rc—RT, but other efficacy metrics have been also described [6]. A statistical test on the observed value of the chosen metrics enables us to conclude on the materiality of the treatment efficacy. The metrics absolute value and its corresponding confidence interval are used to translate the size of the efficacy observed in the trial to a different setting [10].

*How many responders in an RCT*?. Should the experimental treatment be effective, this translates in a true Rc greater than the true RT. Moving forwards, observed values are assumed as the true values. In the experimental treatment arm, there are responders, who encompass the difference in magnitude of effect compared to the control arm. Since all subjects included in the trial are assumed identical, there is a responder counterpart in the control arm. Designated 'potential' responders, unlike responders in the experimental therapy arm, they will not benefit from participating in the trial.

Since all participants are assumed exchangeable, they are in fact identical and we can apply the definition of responders given above. According to that same definition, the number of responders in the experimental therapy arm is given by $d_1$ = (Rc—RT).NT. In fact, they are among the '$d$' participants in the experimental therapy arm who will not develop the event (Table 1). The remaining of the '$d$' patients, $d_2$, are those who would not have presented the event when not receiving the experimental treatment. The total number of responders in the trial, i.e. the number of responders in the experimental treatment arm plus the number of potential responders in the control arm, is (Rc—RT). (Nc + NT).

*"Non-informative" and "unnecessary" participants*. We introduce the concept of unnecessary and non-informative participants in RCTs. Non-informative participants are subjects who will not present the event regardless of being treated or not with the experimental therapy. They are considered not informative with respect to the efficacy measured in the trial as opposed to responders, who carry all the information required to conclude on the treatment's efficacy. On the other hand, unnecessary participants are all the subjects who do not benefit from participating in the trial regarding the prevention of the event, that is to say, non-responders and non-informative participants. Non-responders will present the event whether treated or not with the experimental treatment. These unnecessary participants carry the burden of having to participate in a clinical trial without benefiting from it, which might include experiencing side effects. These various categories are illustrated in Table 3, in the Results section.

*How many non-responders in this RCT*?. According to our definition of responders, the non-responders are those subjects who would experience the event regardless of being administered the experimental treatment or not. Their number in the experimental treatment arm is '$b$' and $a$—(Rc—RT).Nc in the control group. Their total number is $(a + b)$—(Rc—RT).Nc.

*How many non-informative and unnecessary participants are included in a trial*?. Let us deal first with the primary objective of a RCT, which is to generate evidence to support the efficacy, i.e. reduces the rate of the event of interest, of the experimental treatment that happens when the observed difference in rates Rc—RT is large enough to rule out the play of chance. Looking at the figures in Table 1 from a purely theoretical point of view, the gold standard statistical significance can be achieved, among other means, by decreasing '$c$' and '$d$', the number of recruited subjects who will not present the event, whatever the arm they have been randomly allocated to. These '$c + d$' participants do not carry any information regarding the primary objective of the trial. Their main "role" in the trial is to provide a denominator value for computing the rates, Rc and RT. As noted above, the responders in the treatment arm are hidden in '$d$'. These (Rc–RT).NT participants are in effect the only ones to benefit from participating in the trial. Further, they carry the efficacy as captured in the trial. For this reason alone, they must be counted apart. Participants $c + d$—(Rc–RT).NT are said "non-informative" for they

do not contribute with real pharmacologic effect from the experimental treatment. Elaborating on this same reasoning, the non-responders are unnecessary participants for they do not bring into the trial a part of the experimental treatment efficacy and they cannot (when they were allocated to the control arm) or do not (experimental treatment arm) benefit from the treatment. In fact, as shown later by the simulation, a 'perfect' trial does recruit only responders. In total, the number of unnecessary participants in clinical trial is $(c + (d—RT.NT)) + (a + b)—$NR. Hence, we identified and defined four categories of participants in a trial, the last category of which being the sum of the two previous ones (see Table 3): responders, non-responders, non-informative and unnecessary participants, totaling the number of non-responders and non-informative participants.

## Ethical losses and unnecessary participants

A source of recurring dispute within RCTs is their ethical acceptability. For an individual, participating in an RCT can mean different types of ethical loss. For example, exposure to side-effects, loss of access to standard care, or loss of access to the best available and tailored treatment, or other more subtle, if not less detrimental, types of ethical loss such as moral distress or the feeling of being misled by scientists, pharmaceutical companies or even the society. There are, of course, benefits: best care administered for free, chance to be given a new, more effective treatment, to name a few. However, given the diversity of individuals, it is impossible to ensure a positive benefit-loss balance for each participant. There is, therefore, an ethical obligation to minimize the number of participants enrolled for whom the benefit-loss balance would be negative. When the outcome of interest is deemed serious or life-threatening, and death is certainly the most serious one, the marginal benefits, whatever they are, can be disregarded and ethical loss defined as any participant enrolled in the trial who is not a responder. Therefore, the number of ethical losses is $(Nc + NT)—(Rc—RT).NT$.

**A simplified trial summary data.** A total of 7020 participants with type II diabetes were randomized to either the control treatment (C), i.e. placebo, or empagliflozin (T) in the Empa-REG-Outcome trial [11]. In this trial, twice as many participants were randomized to the experimental treatment arm compared to a control arm. For the sake of simplicity, an assumption was made that the experimental treatment had no other effect than preventing death, which was not the case in the actual trial. And, instead of using the composite primary endpoint used in the original trial, mortality from any cause was chosen, which is straightforward in its validity and interpretation, in the Empa-REG trial and facilitates simulations without altering the interpretation of their results. Mortality figures in this simplified version of the original trial are given in Table 2 [11].

**Assumptions.** For the sake of simplicity, a first assumption was made that the probability of death of non-responders was constant. This assumption does not apply in real life for a given disease, the untreated outcome probability varies from one subject to another, thus from one trial to another. Second, the only subgroup that remained the same in our simulations was that of the responders in the experimental treatment arm, i.e. those participants who benefited from having been enrolled in the trial.

Table 2. Mortality figures in the Empa-REG-Outcome Trial [11].

|  | Placebo (C) | Treatment (T) | Total |
|---|---|---|---|
| **Deceased** | 194 | 269 | 463 |
| **Alive** | 2139 | 4418 | 6557 |
| **Total** | 2333 | 4687 | 7020 |

Table 3. Demographic characteristics of participants.

| Categories | Total N in the Empa-REG-outcome trial | Computation | Comments |
|---|---|---|---|
| Responders | 181 | 121 (T group) + 60 (C group) | In the T arm, the number of responders is obtained by subtracting the observed number of deaths (269) from the expected number of deaths if T = C, 390 = 194*(4687/2333); in the C group 60 = 121*(2333/4687) |
| Non-responders | 403 | 269 (T group) + (194–60) (C group) | All deaths in T arm + all deaths in C arm minus the number of responders in the P arm. |
| Non-informative | 6436 | 2139 (number of alive participants in the C group) + 4418 (number of alive participants in the T group)– 121 (responders in the treatment arm) | |
| Total of unnecessary participants | 6839 | 6436 (non-informative) + 403 (non-responders) | This is the total number of participants minus the T responders (121) and potential responders (60) |
| Ethical losses | 6899 | 6839 (total unnecessary participants) + 60 (the potential responders in the C group) | Number equals to total sample size less the number of responders (the "true" responders, those participants who beneficiate from enrolment in the trial) |

**Simulations.** Formulas developed in the previous sections were coded in an Excel spreadsheet and applied to the mortality figures in the Empa-REG-Outcome Trial [11] (Table 2). RCT simulations were carried out varying the proportion of responders. In contrast, as said above, the number of participants in the subgroup of responders in the T arm was kept constant (n = 121, see below in the results section). It was assumed that, for unnecessary participants, the "true" probability of death while untreated is the observed death rate, $R_c$, in the control arm that remained constant across all simulated trials. At each increment of the proportion of responders (0.025), a new RCT arises, and the following values were computed from the new RCT's summary data (Table 1): the number of ethical losses, trial sample size, observed relative risk (RR), and chi-square value for testing drug efficacy. The starting proportion is 1, a value corresponding to the "perfect" trial where all enrolled participants are responders (see below). The final proportion is 0.025, since a proportion that equals 0 results in infinite values. Note that the proportion of responders in the Empa-REG-Outcome trial is 0.023. Computations were done using an Excel spreadsheet. The numerical values of the participants were rounded to the closest unit.

## Results

### How many subjects per category are behind mortality information in the simplified trial?

One of the measures of efficacy in RCTs is the relative risk (RR); in our simplified trial, derived from the Empa-REG-Outcome trial [11], the RR was 0.73. The information on treatment efficacy is entirely carried by the responders. The responders do not appear as such in Table 2, and little derivation is required to bring them off, which is explained below. Other derivations and categories of participants enrolled in the trial are also developed below, and their results are shown in Table 3.

The number of expected deaths in the T group if the participant had not received the experimental treatment is (194*(4687/2333)) = 390 [11]. Thus, the number of responders in the T group is 390–269 = 121, which is another way to compute the number of responders than the one given by definition itself, (Rc—RT).NT. Since the size of the C group is about half the size of the T group, there are potential 60 responders in the C group, who, by definition, are

among the 194 participants in the C group with fatal event, i.e. death. Consequently, a total number of 181 participants (121 + 60) with a responder profile have been recruited in the Empa-REG-Outcome trial, representing only 2.3% of the enrolled participants.

These 181 responders and potential responders have a counterpart, the non-responders. This counterpart is defined as those subjects enrolled in the trial arm T for whom the experimental treatment did not prevent death, i.e. all deceased participants in the T arm (269), plus those participants in the C group for whom the experimental treatment would not have prevented death, i.e. 134 (194–60). Note that participants who stayed alive, whether treated or not, are neither responders nor non-responders, and considered neutral regarding the drug efficacy. The total of unnecessary participants equals 7020–181 = 6839, while the ethical losses (i.e., all the participants who do not benefit from enrolment in the trial regarding the primary outcome) add up to a total 6839 + 60, the potential responders in the C group. The range of participant categories defined against response to drug T is shown in Table 3 below.

In this trial, all those participants who did not benefit from the experimental treatment (i.e. 98.3% of all enrolled subjects) suffered ethical loss (i.e. 7020), minus the number of responders in the T group, 121. Note that the degree of ethical loss depends on the endpoints, with the maximum loss occurring if the endpoint is premature death.

**Relations between the proportion of responders and trial characteristics.** Relations between the percentage of responders and outstanding trial features (i.e., ethical losses, trial sample size, observed relative risk, and chi-square value for testing drug efficacy) are shown in Fig 1 below. All other things being equal, especially the responders and non-informative subject profiles, when the percentage of responders in the trial sample increases from null to 100% (the "perfect" trial sample), the ethical losses, the sample size, thus the overall trial cost, the observed relative risk decrease, while the chi-square value increases. Since the sample size decreases, the overall trial cost decreases, while the chi-square value increases, also increasing the chance of demonstrating the experimental treatment's efficacy. The overall conclusion from these results supports the intuitive conviction that efficacy is less likely demonstrable when responders are insufficiently represented in the trial sample.

**The "perfect" randomized clinical trial.** Were it possible to recruit only responders, this amounts to the perfect RCT: i) all treated participants who otherwise would have died will be alive by the end of the trial; ii) the number of subjects needed to achieve the sample size for randomization would be the lowest possible; iii) the ethical losses minimized. Here the ethical losses amount to 60 (33.4%) of all randomized subjects (Table 4). We notice that even in this "perfect" RCT, there is an inescapable, although minimized, ethical loss.

**"Less perfect" trials.** No matter how much we would like it to happen, identifying responders with 100% accuracy is nothing but a pipe dream. Among the many reasons for this is the simple fact that the drug has not yet been tested in humans in the anticipated context of use in real life, preventing any researcher from inferring the responder profile. However, while work is ongoing to overcome this hurdle and identify responders' profile ahead of the trial as accurately as possible, we explored the consequences of progressively increasing the proportion of responders in our simulated example in Fig 1, showing that ethical losses, the required trial sample size and the relative risk all decrease, while the probability of ascertaining trial significance increases.

## Discussion

The results and their derivation share a series of disruptive points that deserve discussion.

## Panel A
### Percentage of responders and ethical losses

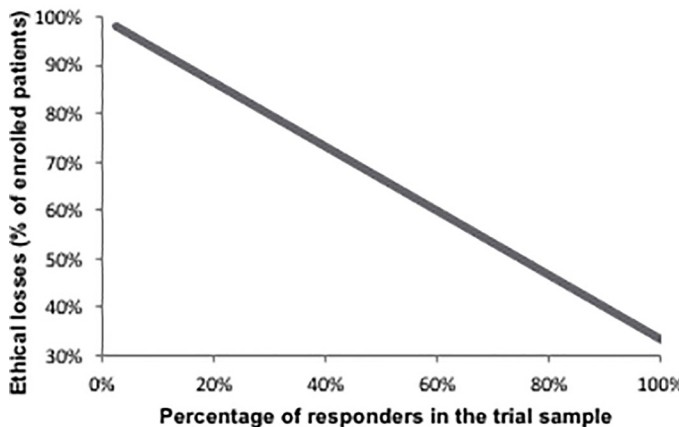

## Panel B
### Percentage of responders and sample size

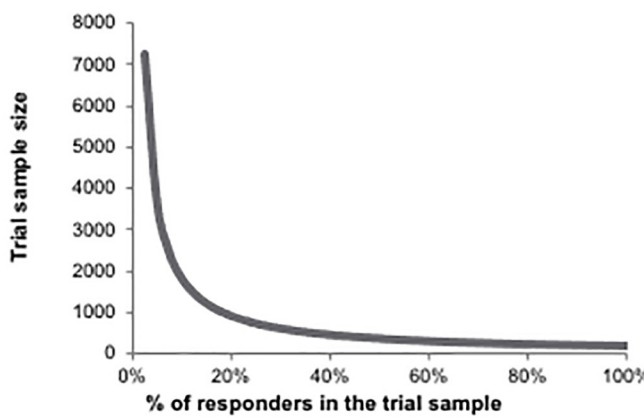

## Panel C
### Percentage of responders and trial relative risk (RR)

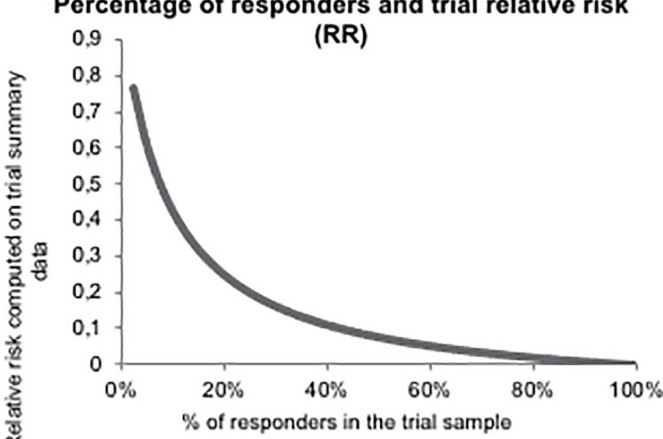

## Panel D
### Percentage of responders and chi-square for the drug efficacy

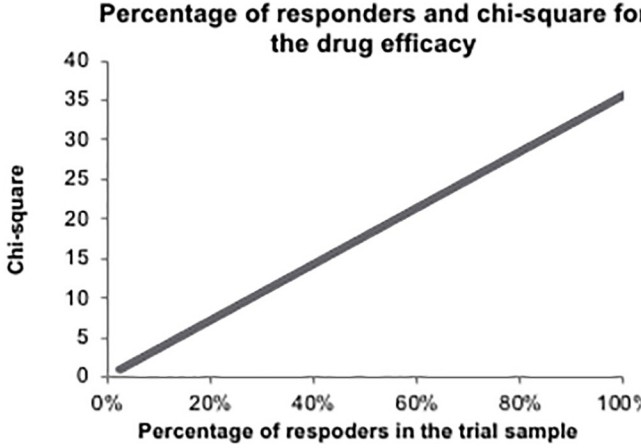

**Fig 1. Relations between the proportion of responders in the trial sample and major features of the trial.** Panel A, shows the linear decrease of the ethical losses expressed in percentage of unnecessary patients with the increase of the percentage of responders. Panel B, keeping the number of responders in the T group constant at n = 121, the sample size drops abruptly from more than 7,000 to less than 1,000 when the responder percentage increases up to about 20% to then flatten until the "perfect" trial status is reached, meaning that only responders are enrolled. Panel C, the relative risk decreases rapidly until the responder percentage reaches about 40%, from less than 0.8 to about 0. Panel D, the chi-square value of the comparison test of mortality rates between placebo and treatment groups increases linearly as the percentage of responders increases. The probability of ascertaining significance increases as well.

## Limitations of this study

Our study has some limitations relating to the theoretical nature of the approach used to conduct the simulations presented here. Could these limitations reduce the impact of the results

**Table 4. Ethical losses.**

|          | C   | T   |     |
|----------|-----|-----|-----|
| Deceased | *60* | *0* | 60  |
| Alive    | *0*  | *121* | 121 |
|          | 60  | 121 | 181 |

obtained? In a real trial, participants are not interchangeable; they may withdraw or disappear, may take the treatment incorrectly, and may have been wrongly classified as eligible. For simplicity, it was decided to dismiss these potential limitations. Taking into account the non-compliance with the trial procedures and treatment would lead to different figures in Table 3 but would not change the direction of the results. Moreover, we have made it clear from the outset that the study objective was of qualitative and not quantitative nature.

The impact of the participant interchangeability hypothesis is more difficult to assess. Imagine that the individuals included in the trial are grouped into clusters of similar characteristics. Within a cluster, participants are interchangeable. From one cluster to another, the proportion of responders might vary. If we apply the same approach to each cluster, the overall result will remain qualitatively the same.

## Ambiguity of clinical trial objectives

A trial said to be "aimed at demonstrating the efficacy of X on preventing death in patients with type 2 diabetes", has two intertwined meanings. First, the trial is expected to show that X is more effective than the control, and second, that it will provide an estimate of the effect size. We have shown that observed efficacy in the trial and therefore the chance to demonstrate efficacy is dependent on the proportion of responders, whereas the genuine efficacy of the experimental treatment is a property that is independent of this proportion.

## Is treatment efficacy carried by responders in full?

Traditionally, response to treatment and the treatment effect size are the cornerstones of efficacy demonstration for any RCT. Such assumptions have two major consequences. First, efficacy is diluted across the entire sample of participants. By applying the responder paradigm, as shown here, efficacy is no longer diluted. For the recruited subjects, the tested treatment is or is not effective, it is a dichotomous (yes/no) response. While such a way of looking at efficacy is acceptable for a binary event such as death or relapse of HBV infection, for example, it seems less relevant for a continuous outcome such as pain. However, even pain can be binarized if we take into account each participant's own threshold of pain acceptability. Second, translation to other populations is often achieved by applying the measured relative risk. With this responder paradigm, efficacy indices obtained on one population are irrelevant to another population (see panel C, Fig 1) because it is likely that the fraction of responders will vary between populations, and so will the relative risk and all other indices. The point here is that the proportion of subjects not at risk and responders randomized in either arm will change, and thus the relative risk will change accordingly. This is a consequence of the Effect Model (EM) law, a principle that has been verified both empirically and theoretically [12]. The EM law says that the absolute benefit in a population varies as a function of individual or subgroup rate of events ($R_c$), whatever is the treatment of interest. Therefore, both $R_t$ and $R_c$ vary, and so does the relative risk. The relative risk remains constant in one instance only: when the EM is linear, which seems quite infrequent.

## Non-informative and/or unnecessary participant = ethical loss; how does the equation fare?

Central in the argumentation is the idea that data on efficacy in an RCT is only carried by the responders. This proportion of responders is the taproot of the ethical losses since, for those who are not responders, the burden of participating in a trial is not offset by the benefit expected.

### Is focusing on the unnecessary trial participants a way to address ethical issues in an RCT?

As exposed in the introduction, the uncertainty raised from enrolling individuals in an RCT has two opposite uncertainties. On the one hand, the standard treatment is possibly less effective or more harmful than the experimental treatment. On the other, the standard treatment is known to be effective, whereas the efficacy of that in the experimental arm is postulated. To help handle the two opposite uncertainties, Freedman proposed the concept of "equipoise" [1]. An investigator can enroll subjects in an RCT if there is genuine uncertainty about the preferable treatment [13]. It is interesting to note that in a review of childhood oncology trials, the odds ratio for survival was 0.96, suggesting that the equipoise has been respected [14]. This finding points out to a limitation of this equipoise principle: is it ethical to launch a trial if the evidence in favor of an expected superior efficacy of the experimental treatment against the control is almost nil? Our results show that identifying responders would in fact reduce uncertainty and can be viewed as an operational equipoise principle.

### Is the "perfect" RCT "fully" ethical?

When we considered the "perfect" RCT, with 100% of responders, we noted that the 60 participants randomized to the control arm, had all deceased by the end of the trial: the ethical loss for this "perfect" RCT was 33%. Thus, even a perfect randomized trial cannot annul the number of unnecessary participants. This fact deserves three comments: i) the ethical loss can be reduced by lowering the fraction of subjects randomized into the control arm; ii) however, the ethical loss in a RCT cannot be avoided; iii) increasing the proportion of responders implies that a reliable tool is required to enable their selection.

Other ethical issues remain for responders such as the amount and type (e.g., invasive investigations) of data collected, number of investigation time-points, confidentiality, and informed consent process.

### Selection of responders for enrolment in RCT: Neither new nor easy

We demonstrated that selecting responders in RCT is of benefit to overcome ethical limitations of RCTs but also to address other practical issues, such as the trial sample size. Sequential designs were long proposed to reduce the number of subjects enrolled [15,16]. Increasing the proportion of responders is one of the arguments in favor of adaptive designs [5]. But the issue remains, how to select responders ahead of trial onset or more often during the trial. Several ways have been explored: responders identified with markers of response [17], good compliers as assessed in a run-in phase [18], play-the-winner [19], discontinuation for non-responders or considerably improved participants in a run-in phase [20], post-hoc selection, for instance, non-responders discontinuation trial designs [21]. All these attempts have not been widely adopted either because they were not proven successful, or because they are difficult to implement or because they were considered unethical [22].

A tool that can identify responders before enrolment is clearly needed. More specifically, the false positive error rate must be 0. We are still far from achieving fully reliable tools, although potential solutions as to how they could be designed are now available [23]. Nevertheless, if this is achieved, the permanence of RCT as the gold standard will be more strongly and more rightly challenged. Most importantly, the design and undertaking of RCTs will have to change as a result. The availability of tools for selecting responders will raise two challenging issues: the validation and implementation of responder selection. The reduced sample size is an additional consequence to bear in mind. Further, should such a tool be successfully

designed, this would undoubtedly change the applicability of the concept of personalized medicine concept as it is currently known.

## Conclusion

The simulation reported here confirms that increasing the proportion of responders in a RCT is worthwhile on ethical grounds, mainly because it reduces the number of subjects enrolled who do not provide information on treatment efficacy.

We claim that the low proportion of responders is parallel to and a marker of the uncertainty relating to the experimental treatment efficacy ahead of trialing new drugs. Scannel and Bosley have shown that, currently, drug development processes are poorly informative [24]. According to new policies issued by the FDA and EMA, performing *in silico* clinical trials might be a solution to improve the efficiency of R&D by reducing uncertainty across all the stages [25–27]. Individualized computer simulations could become therefore a valuable tool to identify responders ahead of designing RCT [12,28].

## Supporting information

**S1 File. Ethical losses responders computations.**
(XLSX)

## Acknowledgments

The authors thank Rita Moreira da Silva and Paulo Pacheco for their contribution to the editing and formatting of the manuscript.

## Author Contributions

**Conceptualization:** Jean-Pierre Boissel.

**Data curation:** Jean-Pierre Boissel.

**Formal analysis:** Jean-Pierre Boissel.

**Funding acquisition:** Marc Hommel.

**Methodology:** Jean-Pierre Boissel.

**Project administration:** Jean-Pierre Boissel, Ingrid Klingmann, Marc Hommel.

**Software:** Jean-Pierre Boissel.

**Supervision:** Jean-Pierre Boissel.

**Validation:** Jean-Pierre Boissel, David Pérol, Hervé Décousus, Ingrid Klingmann, Marc Hommel.

**Visualization:** Jean-Pierre Boissel, David Pérol, Hervé Décousus, Ingrid Klingmann, Marc Hommel.

**Writing – original draft:** Jean-Pierre Boissel.

**Writing – review & editing:** Jean-Pierre Boissel, David Pérol, Hervé Décousus, Ingrid Klingmann, Marc Hommel.

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
