## [Decision Letter · Decision Letter 0]

21 May 2021

PONE-D-21-01394

Using numerical modelling and simulation to assess the ethical burden in clinical trials and how it relates to the proportion of responders in a trial sample

PLOS ONE

Dear Dr. Boissel,

Thank you for submitting your manuscript to PLOS ONE. After careful consideration, we feel that it has merit but does not fully meet PLOS ONE’s publication criteria as it currently stands. Therefore, we invite you to submit a revised version of the manuscript that addresses the points raised during the review process.

ACADEMIC EDITOR: As per previous comments.

We look forward to receiving your revised manuscript.

Kind regards,

Dylan A Mordaunt

Academic Editor

PLOS ONE

Additional Editor Comments:

Thank you for your submission. The reviewers have found some interest in this, however also some significant suggestions for consideration. Quantitation and modelling of ethical considerations has value insofar as the ability to operationalize ethical considerations in decision-making frameworks. I think there's value in revision and resubmission, albeit the suggestions are not insubstantial.

Journal Requirements:

3. Thank you for stating the following in the Funding Statement Section of your manuscript:

[This work was supported by RESSTORE project (www.resstore.eu) funded by the European Commission under the H2020 program (Grant Number 681044).Authors are independent of the funder.]

 [The funders had no role in study design, data collection and analysis, decision to publish, or preparation of the manuscript.]

Reviewers' comments:

Reviewer's Responses to Questions

**Comments to the Author**

1. Is the manuscript technically sound, and do the data support the conclusions?

Reviewer #1: Partly

Reviewer #2: Yes

2. Has the statistical analysis been performed appropriately and rigorously? 

Reviewer #1: Yes

Reviewer #2: Yes

3. Have the authors made all data underlying the findings in their manuscript fully available?

Reviewer #1: No

Reviewer #2: Yes

4. Is the manuscript presented in an intelligible fashion and written in standard English?

Reviewer #1: Yes

Reviewer #2: Yes

5. Review Comments to the Author

Reviewer #1: PONE-D-21-01394

Using numerical modelling and simulation to assess the ethical burden in clinical trials and how it relates to the proportion of responders in a trial sample

In the present analysis the authors analyse the number of participants within a trial who failed to have an event (death) within a 2-armed parallel RCT. They argue that a proportion of participants fail to benefit from participation on the basis that they never would have had the event in question, and that removing these individuals from trials, such that only responders are included would minimise ethical losses. The authors illustrate this with data from the Empa-REG-Outcome trial.

While this is perhaps a nice example as a statistical exercise, the challenge is what to do with the paper. The takeaway – we should try and have trials that are 100% responders – does not need the mathematical explanation. If we knew that, we wouldn’t need a trial we could just develop a drug and then give it the people we knew would respond.

Instead, the reality is that we do not know who will respond, and if the drug is given to a patient population (note this is the frame upon which the trial must be based) then there will likely be some group – based on some identified characteristic – that who benefits and some that does not.

I also have some specific comments:

1.The concept of ethical losses is not fleshed out; the manuscript would benefit from a clearer description of this concept. This is important given that later (p9lines 179,182) the authors talk about different types of ethical loss.

2.The authors state that “Despite their wide acceptance, questions have been raised about the ethics of randomizing participants to conventional treatments when deemed less effective, to placebo, or to an inadequate treatment if the therapy tested turns out to be less effective or potentially harmful.” – each of these have a specific context that the current phrasing excludes. For example, the initial component of the sentence likely relates to disruption of equipoise about the treatments; concerns about randomisation to placebo largely attend when treatment is available and so reflects the withholding of know treatment, and the third appears to be a post hoc determination.

3.The line about a different setting seems somewhat irrelevant to the manuscript.

4.I would disagree that consent is seen as a global answer to the above issues; but rather is a necessary component to conveying the state of knowledge. I don’t think anyone would suggest that a placebo becomes acceptable on the basis of consent. The acceptability of the placebo is an independent consideration.

5.On p3, lines 64 – 70, the authors write: “It is known that when a new treatment is being trialed, some patients will present the event the treatment is supposed to prevent as opposed to the “responders” who will show the expected reaction to treatment. Identification of responders is a timely and challenging issue that remains poorly studied, mainly because it is difficult to predict who responders and non-responders will be. But beyond the importance of identifying responders in the era of personalized medicine, the consequences of enrolling non-responders in RCTs remain unexplored.” – If this is the motivation, the discussion doesn’t reflect this. What is the importance with respect to personalised medicine (would the identification not be post-hoc analysis to try and find some causally related characteristic?)

6.On lines 72-78 the authors write “More precisely, the objective of our work is to show how can RCT simulations be used to address the consequences on the researcher’s ethical obligation of diluting the responders in the trial sample. Our simulations do not aim at quantitatively defining adequate parameters of an RCT but intend to draw upon the qualitative perception of the issue of ethical loss linked to the current way of RCTs are planned, especially when it comes to eligibility criteria. To that extent, our approach focuses on treatment efficacy.”

a.What is the ethical obligation of diluting the responders in the trial sample? I am not clear what this is referring to.

b.I am not sure how the work addresses the consequences of the analysis.

Methods

Given the extensive exposition of the formulas, why were they not used? It seems strange to spend the time detailing the approach, only to then not use it and use an approach that isn’t described.

The terminology changes such that P is used for placebo, then the subscript c (for control?) is subsequently used.

The lumping together of the group d is confusing. It would have been more intuitive to have these designated as two groups d1 and d2 to more explicitly note that d1 are those alive who would never have had the event, and those d2 those that would have had the event but benefited from the treatment.

On p8, lines 157-162, the authors state: “Looking at the figures in Table 1 from a purely arithmetic point of view, the gold standard statistical significance can be achieved, among other means, by decreasing ‘c’ and ‘d’, the number of recruited subjects who will not present the event, whatever the arm they have been randomly allocated to. These ‘c + d’ participants do not carry any information regarding the primary objective of the trial.” – this could do with greater exposition abut how this is achieved, at present this is left unclear.

Can the authors explain more why mortality trumps all other benefits such that they can be disregarded – this claim surely needs to be better justified? In terminal or palliative care, for example, a drug which may not prolong life or stave off death may be beneficial if it sufficiently advances quality of life (one may contend).

Indeed, the simplicity of the analysis does not reflect reality and the other outcomes of interest (indeed in the authors own trial the outcome was a composite outcome indicating that there was at least some suggestion that other outcomes may be as important.

Doesn’t the assumption about probability of death mask the complexity of identifying non-responders and responders? Again, this goes to the utility of the analysis – if the assumptions are not valid, how should the results be taken?

Reviewer #2: Summary: This manuscript reports on a complex hypothetical study which uses statistical and numerical modelling in an attempt to map out the proportion of ‘responders’ and ‘non responders’ in randomized control trials (RCTs) for comparison of new drugs or treatments, in an attempt to reduce the number of ‘non-informative participants’ or ‘unnecessary’ participants in RCTs. By identifying such non-responders in RCTs, the authors argue that this would help to minimize ‘ethical losses’ characterized for example as “exposure to side-effects, loss of access to standard care, or loss of access to 182 the best available and tailored treatment.”

While he manuscript is original ad attempts to resolve some of the ethical and moral dilemmas associated with the RCT format for investigational drugs or therapy. The manuscript as currently written is not easily comprehensible to the ‘non-expert’ or general reader and will require some modifications to enhance clarity and reduce some identified ambiguities in the current manuscript prior to acceptance for publication.

Suggested areas for further clarification and improvement of the manuscript:

1.a). In the Abstract the authors appear to use the following terms interchangeably i.e., “non-responders,”; “unnecessary participants” and “non-informative participants” …. The question that comes to mind is this…. Do these terms refer to the same fragment of the sample population, or are they different in anyway?

b). Again, on Page 8, Lines 176-177, the authors state “non-responders, non-informative and unnecessary participants, totaling (sp) the number of non-responders and non-informative participants.”

c). Furthermore, on Page 8, Lines 168-171, the authors state “Elaborating on this same reasoning, the non-responders are unnecessary participants for they do not bring into the trial a part of the experimental treatment efficacy and they cannot (when they were allocated to the control arm) or do not (experimental treatment arm) benefit from the treatment.”

One would like to suggest that this apparent use of these different terminologies of ‘non-responder’, ‘unnecessary participants’ and ‘non-informative participants’ creates a confusion for the reader who has to try to decipher the meaning of each of this categories…I therefore suggest that the authors should clearly define each of these categories…Are they the same or are they different?...If they are the same then say so…if they are different , then clearly define how they are different from each other? Are some of these categories essential to the RCT format…if so, which ones are essential or non-essential?... The way to approach this is suggested in 2. Below:

2.On page 5, Line 104…the authors have inserted a subheading for “Responders and Non-Responders”. However, after including a section “Definition of Responders” and clearly elaborating on “Responders”, the reader is left in the dark about a definition for “Non-Responders”? Therefore, following on from the questions raised in Paragraphs 1a, b, c above…One would like to suggest that the authors should clearly define who the ‘Non-responders’ in this study are, and also to indicate whether they are the same as ‘unnecessary participants’ and ‘non-informative participants…. This will assist in enhancing the clarity and comprehension of the arguments raised in the rest of the article/manuscript….

3.On page 9 Lines 178-182, the authors have added a subsection on “Ethical losses and unnecessary participants”. Here the authors state, “A source of recurring dispute within RCTs is their ethical acceptability. For an individual, participating in an RCT can mean different types of ethical loss, for example, exposure to side-effects, loss of access to standard care, or loss of access to the best available and tailored treatment.”. One would like o suggest that other important potential ethical losses which ought to be discussed would be issues of ‘moral distress and justice’…For example where individuals have volunteered for an RCT with the hope of gaining a potential benefit or even contributing to science only to find out that there their participation was perchance ‘unnecessary’…This can create some form of ‘moral distress’ and a reluctance to participate in future RCTs. Furthermore, with regards to justice, where individuals have invested time and effort or money e.g., taking time off work to participate in an RCT, it could be considered an injustice or unjust and unethical to use peoples time and effort without any justifiable benefits either for themselves or to society in general…. This would be the case regardless of whether the individual subjects participation was covered under the informed consent doctrine? …

4.On pages 17-18, Lines 333-345, The authors have described the concept of ‘equipoise in RCTs’. One would like to suggest that moving this section to the ‘Introduction’ section from the ‘Discussion’ which could serve as a background to RCT studies and provide a justification on why it would be important to identify “responders” and “non-responders” in RCTs….??

5.Insert a statement of limitations which identifies potential limitations to assumptions made in this study…For example are there any other confounding variables which could impact on the assumptions made for this study? E, g. What about participants who withdraw prematurely from an RCT... How will this impact on the percentage or proportion of ‘responders’ and ‘non-responders’…. Because the assumptions illustrated in this case are based on the principle of ‘all things being equal’…??

6.Finally, the statistical parameters and calculations for this study should be further reviewed by another qualified biostatistician

Minor Corrections: Few typographical and grammatical errors…

6. PLOS authors have the option to publish the peer review history of their article (what does this mean?). If published, this will include your full peer review and any attached files.

Reviewer #1: No

Reviewer #2: **Yes: **Sylvester C. Chima, MD, LLM, LLD

---

## [Author Response · Author response to Decision Letter 0]

21 Jul 2021

Reviewer(s)' Comments to Author

REVIEWER #1

Comments to Author: In the present analysis the authors analyse the number of participants within a trial who failed to have an event (death) within a 2-armed parallel RCT. They argue that a proportion of participants fail to benefit from participation on the basis that they never would have had the event in question, and that removing these individuals from trials, such that only responders are included would minimise ethical losses. The authors illustrate this with data from the Empa-REG-Outcome trial.

While this is perhaps a nice example as a statistical exercise, the challenge is what to do with the paper. The takeaway – we should try and have trials that are 100% responders – does not need the mathematical explanation. If we knew that, we wouldn’t need a trial we could just develop a drug and then give it the people we knew would respond.

Instead, the reality is that we do not know who will respond, and if the drug is given to a patient population (note this is the frame upon which the trial must be based) then there will likely be some group – based on some identified characteristic – that who benefits and some that does not.

1. The concept of ethical losses is not fleshed out; the manuscript would benefit from a clearer description of this concept. This is important given that later (p9lines 179,182) the authors talk about different types of ethical loss.

Authors’ reply: Change 1. Ethical losses in randomized controlled clinical trials (RCTs) is an intuitive and rather vague concept that can be outlined at a first glance as a loss of chance, everything a patient either misses or acquires as detrimental for participating in trials.

“In order to propose a more precise definition and explore how to reduce ethical losses in randomized controlled clinical trials (RCTs), we set out to identify trial participants who do not contribute to demonstrating that the treatment in the experimental arm is superior to that in the control arm. RCTs emerged mid-last century as the gold standard for assessing efficacy, becoming the cornerstone of the value of new therapies, yet their ethical grounds are a matter of debate.”

2. The authors state that “Despite their wide acceptance, questions have been raised about the ethics of randomizing participants to conventional treatments when deemed less effective, to placebo, or to an inadequate treatment if the therapy tested turns out to be less effective or potentially harmful.” – each of these have a specific context that the current phrasing excludes. For example, the initial component of the sentence likely relates to disruption of equipoise about the treatments; concerns about randomisation to placebo largely attend when treatment is available and so reflects the withholding of know treatment, and the third appears to be a post hoc determination.

Authors’ reply: change 2 There are various types of ethical concerns linked with randomization: prior to launch the trial of a new treatment, randomizing participants to conventional treatments that is deemed less effective is a breach in the equipoise principle, as allocating them to placebo, or to a known sham treatment; once the trial is achieved, if the therapy tested turns out to be more effective than the conventional one, or less effective or potentially harmful, participants in the control arm in the former case or those in the experimental arm in the latter were faced a loss of chance.

3. The line about a different setting seems somewhat irrelevant to the manuscript.

Authors’ reply: the sentence has been removed

4. I would disagree that consent is seen as a global answer to the above issues; but rather is a necessary component to conveying the state of knowledge. I don’t think anyone would suggest that a placebo becomes acceptable on the basis of consent. The acceptability of the placebo is an independent consideration.

Authors’ reply: We agree. The statement is too blunt, and, as such, erroneous. 

Change 4:... way to deal �3� with these issues, but it transfers concerns from investigators to participants.

5. On p3, lines 64 – 70, the authors write: “It is known that when a new treatment is being trialed, some patients will present the event the treatment is supposed to prevent as opposed to the “responders” who will show the expected reaction to treatment. Identification of responders is a timely and challenging issue that remains poorly studied, mainly because it is difficult to predict who responders and non-responders will be. But beyond the importance of identifying responders in the era of personalized medicine, the consequences of enrolling non-responders in RCTs remain unexplored.” – If this is the motivation, the discussion doesn’t reflect this. What is the importance with respect to personalised medicine (would the identification not be post-hoc analysis to try and find some causally related characteristic?) 

Authors’ reply: We acknowledge that, although we demonstrated its interest, the consequences of selecting responders are not developed. We think this should better be the topic of another paper, or editorial. We wanted to focus on the demonstration. However, (changes 17 and 18) we added allusions to these consequences.

6. On lines 72-78 the authors write “More precisely, the objective of our work is to show how can RCT simulations be used to address the consequences on the researcher’s ethical obligation of diluting the responders in the trial sample. Our simulations do not aim at quantitatively defining adequate parameters of an RCT but intend to draw upon the qualitative perception of the issue of ethical loss linked to the current way of RCTs are planned, especially when it comes to eligibility criteria. To that extent, our approach focuses on treatment efficacy.”

a. What is the ethical obligation of diluting the responders in the trial sample? I am not clear what this is referring to.

b. I am not sure how the work addresses the consequences of the analysis.

Authors’ reply: Reviewer point “a”. The sentence is misleading. Change 5: consequences of diluting the responders in the trial sample.

“on the researcher’s ethical obligation” was removed from the manuscript.

Reviewer point “b”. Our work does not address the consequences of the trial data analysis.

7. Given the extensive exposition of the formulas, why were they not used? It seems strange to spend the time detailing the approach, only to then not use it and use an approach that isn’t described.

Authors’ reply: A clear lack of explanation. The approach was detailed with the formulas since it was the core of the simulations. Change 6: Formulas developed in the previous sections were coded in an Excel spreadsheet and applied to the mortality figures in the Empa-REG-Outcome Trial �11� (Table 2).

8. The terminology changes such that P is used for placebo, then the subscript c (for control?) is subsequently used.

Authors’ reply: Thank you for picking this up. We have changed placebo (P) for control (C) throughout the revised manuscript.

9. The lumping together of the group d is confusing. It would have been more intuitive to have these designated as two groups d1 and d2 to more explicitly note that d1 are those alive who would never have had the event, and those d2 those that would have had the event but benefited from the treatment.

Authors’ reply: We take the point and proceed to differentiate d1 and d2 (change 7) throughout the manuscript (actually in the reverse order, in line with the order of appearance in the text, d1 being the responders in the d arm, d2 being those in the d arm who would not have the event if not treated)

10. On p8, lines 157-162, the authors state: “Looking at the figures in Table 1 from a purely arithmetic point of view, the gold standard statistical significance can be achieved, among other means, by decreasing ‘c’ and ‘d’, the number of recruited subjects who will not present the event, whatever the arm they have been randomly allocated to. These ‘c + d’ participants do not carry any information regarding the primary objective of the trial.” – this could do with greater exposition about how this is achieved, at present this is left unclear.

Authors’ reply: In fact, the operation consisting in decreasing “c” and “d” is purely theoretical and has no practical counterpart. It is detailed here to stress that the “no event” participants’ role is mostly to enhance statistical power. We agree that the wording is not clear enough. The reviewer point is considered by Change 8: “arithmetic” is replaced by “theoretical”.

11. Can the authors explain more why mortality trumps all other benefits such that they can be disregarded – this claim surely needs to be better justified? In terminal or palliative care, for example, a drug which may not prolong life or stave off death may be beneficial if it sufficiently advances quality of life (one may contend). Indeed, the simplicity of the analysis does not reflect reality and the other outcomes of interest (indeed in the authors’ own trial the outcome was a composite outcome indicating that there was at least some suggestion that other outcomes may be as important).

Authors’ reply: We fully agree with the reviewer. We choose mortality for the simulation because the number of deaths in the Empa trial does not suffer any ambiguous assessment. We wrote “And, instead of using the composite primary endpoint used in the original trial, mortality from any cause was chosen, which is straightforward in its validity and interpretation.” This is by no means a general statement on what matters to real patients. In order to make it clearer, we added “in the Empa-REG trial and facilitate simulations without altering the interpretation of their results” (change 9)

12. Doesn’t the assumption about probability of death mask the complexity of identifying non-responders and responders? Again, this goes to the utility of the analysis – if the assumptions are not valid, how should the results be taken?

Authors’ reply: As stressed in change 10, it is hardly possible today to identify responders prior to launching an RCT. For us, that does reduce the interest of assessing the role of the proportion of responders in the ethical losses in a RCT. It may be true that, once appropriate tools are available, identifying responders (vs non-responders) to a mortality-decreasing treatment would be easier than for another outcome. However, since it is not the issue in this work, we deemed it not necessary to elaborate on this issue. In order to make it clearer, we modified the sentence (change 10) “this definition is to day difficult, if not impossible, to apply: how to predict that a given patient is a responder?”

We did not mention the post-hoc identification which, because of the error rate adjustment required by the statistical dragging of the data, does not result in reliable findings. Here, the issue is rather prior identification.

REVIEWER #2

Comments to Author: This manuscript reports on a complex hypothetical study which uses statistical and numerical modelling in an attempt to map out the proportion of ‘responders’ and ‘non responders’ in randomized control trials (RCTs) for comparison of new drugs or treatments, in an attempt to reduce the number of ‘non-informative participants’ or ‘unnecessary’ participants in RCTs. By identifying such non-responders in RCTs, the authors argue that this would help to minimize ‘ethical losses’ characterized for example as “exposure to side-effects, loss of access to standard care, or loss of access to 182 the best available and tailored treatment.”

While he manuscript is original ad attempts to resolve some of the ethical and moral dilemmas associated with the RCT format for investigational drugs or therapy. The manuscript as currently written is not easily comprehensible to the ‘non-expert’ or general reader and will require some modifications to enhance clarity and reduce some identified ambiguities in the current manuscript prior to acceptance for publication.

Suggested areas for further clarification and improvement of the manuscript:

1. a) In the Abstract the authors appear to use the following terms interchangeably i.e., “non-responders,”; “unnecessary participants” and “non-informative participants” …. The question that comes to mind is this…. Do these terms refer to the same fragment of the sample population, or are they different in anyway?

b) Again, on Page 8, Lines 176-177, the authors state “non-responders, non-informative and unnecessary participants, totaling (sp) the number of non-responders and non-informative participants.”

c) Furthermore, on Page 8, Lines 168-171, the authors state “Elaborating on this same reasoning, the non-responders are unnecessary participants for they do not bring into the trial a part of the experimental treatment efficacy and they cannot (when they were allocated to the control arm) or do not (experimental treatment arm) benefit from the treatment.”

One would like to suggest that this apparent use of these different terminologies of ‘non-responder’, ‘unnecessary participants’ and ‘non-informative participants’ creates a confusion for the reader who has to try to decipher the meaning of each of this categories…I therefore suggest that the authors should clearly define each of these categories…Are they the same or are they different?...If they are the same then say so…if they are different , then clearly define how they are different from each other? Are some of these categories essential to the RCT format…if so, which ones are essential or non-essential?... The way to approach this is suggested in 2.

Authors’ reply: We agree that the way we used the three words in the Abstract and even in the core text is confusing. In fact, the three words do not cover the same reality, as illustrated in Table 3 in the Results section. We try to make it clearer in the Abstract (change 11) and in the text (change 12)

In short:

• non-informative participants = patients who would not have the event whether administered or not the experimental treatment 

• non-responders = patients who would present the event with or without the experimental treatment

• unnecessary participants: all the patients who do not benefit from participating in the trial regarding the prevention of the event, that is to say, non-responders + non-informative participants

2. On page 5, Line 104…the authors have inserted a subheading for “Responders and Non-Responders”. However, after including a section “Definition of Responders” and clearly elaborating on “Responders”, the reader is left in the dark about a definition for “Non-Responders”? Therefore, following on from the questions raised in Paragraphs 1a, b, c above… One would like to suggest that the authors should clearly define who the ‘Non-responders’ in this study are, and also to indicate whether they are the same as ‘unnecessary participants’ and ‘non-informative participants…. This will assist in enhancing the clarity and comprehension of the arguments raised in the rest of the article/manuscript.

Authors’ reply: See above. Following the advice of Reviewer 2, we added a section to define each category (change 12).

3. On page 9 Lines 178-182, the authors have added a subsection on “Ethical losses and unnecessary participants”. Here the authors state, “A source of recurring dispute within RCTs is their ethical acceptability. For an individual, participating in an RCT can mean different types of ethical loss, for example, exposure to side-effects, loss of access to standard care, or loss of access to the best available and tailored treatment.”. One would like o suggest that other important potential ethical losses which ought to be discussed would be issues of ‘moral distress and justice’…For example where individuals have volunteered for an RCT with the hope of gaining a potential benefit or even contributing to science only to find out that there their participation was perchance ‘unnecessary’…This can create some form of ‘moral distress’ and a reluctance to participate in future RCTs. Furthermore, with regards to justice, where individuals have invested time and effort or money e.g., taking time off work to participate in an RCT, it could be considered an injustice or unjust and unethical to use peoples time and effort without any justifiable benefits either for themselves or to society in general…. This would be the case regardless of whether the individual subject’s participation was covered under the informed consent doctrine.

Authors’ reply: We fully agree that there are other types of potential ethical losses as a result of participating in an RCT and we thank the Reviewer for raising this issue. And partially 1) the moral distress and 2) the feeling of having been cheated. 

We added these others types of losses (change 13). However, we did not elaborate further since describing the ethical losses in this setting is not the objective of this work.

4. On pages 17-18, Lines 333-345, the authors have described the concept of ‘equipoise in RCTs’. One would like to suggest that moving this section to the ‘Introduction’ section from the ‘Discussion’ which could serve as a background to RCT studies and provide a justification on why it would be important to identify “responders” and “non-responders” in RCTs….??

Authors’ reply: Agree. In fact, when drafting the paper, we hesitated to develop further the concept of ‘equipoise’ in the introduction. In answer to this request, we have moved on the part of the discussion on ‘equipoise’ to the ‘introduction’ (change 14). However, we believe we should also raise the issue of ‘equipoise’ in the ‘discussion’ section of the revised version to stress that our results can help to make the ‘equipoise’ principle more workable.

5. Insert a statement of limitations which identifies potential limitations to assumptions made in this study…For example are there any other confounding variables which could impact on the assumptions made for this study? E, g. What about participants who withdraw prematurely from an RCT... How will this impact on the percentage or proportion of ‘responders’ and ‘non-responders’…. Because the assumptions illustrated in this case are based on the principle of ‘all things being equal’…??

Authors’ reply: Thank you for raising this very good point. We added a section on limitations in the ‘discussion’ (change 15)

6. Finally, the statistical parameters and calculations for this study should be further reviewed by another qualified biostatistician

Authors’ reply: We believe this is a request for the Editorial Team.

7. Minor Corrections: Few typographical and grammatical errors…

Authors’ reply: Typos and grammatical errors were addressed as requested.

---

## [Decision Letter · Decision Letter 1]

21 Sep 2021

Using numerical modeling and simulation to assess the ethical burden in clinical trials and how it relates to the proportion of responders in a trial sample

PONE-D-21-01394R1

Bonjour Professor Boissel,

We’re pleased to inform you that your manuscript has been judged scientifically suitable for publication and will be formally accepted for publication once it meets all outstanding technical requirements.

Merci beaucoup,

Dylan A Mordaunt

Academic Editor

PLOS ONE

Additional Editor Comments (optional):

Reviewers' comments:

Reviewer's Responses to Questions

**Comments to the Author**

1. If the authors have adequately addressed your comments raised in a previous round of review and you feel that this manuscript is now acceptable for publication, you may indicate that here to bypass the “Comments to the Author” section, enter your conflict of interest statement in the “Confidential to Editor” section, and submit your "Accept" recommendation.

Reviewer #2: All comments have been addressed

Reviewer #3: (No Response)

2. Is the manuscript technically sound, and do the data support the conclusions?

Reviewer #2: Yes

Reviewer #3: Yes

3. Has the statistical analysis been performed appropriately and rigorously? 

Reviewer #2: Yes

Reviewer #3: I Don't Know

4. Have the authors made all data underlying the findings in their manuscript fully available?

Reviewer #2: Yes

Reviewer #3: Yes

5. Is the manuscript presented in an intelligible fashion and written in standard English?

Reviewer #2: Yes

Reviewer #3: No

6. Review Comments to the Author

Reviewer #2: (No Response)

Reviewer #3: This paper uses modeling to demonstrate the effect of increasing the number of responders on ‘ethical loss,’ i.e., the number of individuals for whom participating in research is ‘unnecessary,’ as they are either non-responders or because their participation is non-informative. Intuitively, they find that as the number of responders in a trial increases, the number of unnecessary participants decreases (as does sample size) and trial power increases.

As these results are intuitive, I am struggling to see what this analysis adds and encourage the authors to focus their time and attention on the ‘so what’ of this paper. We know non-responders are a problem for RCTs. We also know that it is not possible to identify non-responders a priori. What then is the authors’ takeaway message?

I have made a number of suggestions for further clarification and improvement of the manuscript:

1. Line 50 I suggest replacing the phrase “loss of chance” with “loss of opportunity” throughout the manuscript.

2. Line 49-54 I am unclear about the link that is being drawn between the concept of ‘ethical loss’ and equipoise and suggest that reorganization of the introduction would be helpful. Namely, I suggest leading with a discussion of why RCTs are ethically fraught, followed by the issue of responders vs. non-responders and their effect on trials, and finally I would introduce the concept of ‘ethical loss’ and what the paper is trying to do.

Along those lines, I think it would be important to provide more substantive material on how non-responders dilute trials and the statistical difficulties such dilution produces. As it is currently written, I don’t think there is enough background information for a general reader to easily understand the material.

3. Line 116-118 The authors state that they will provide two examples of why it is important to separate the qualitative objectives of RCTs from the quantitative objectives and then only provide one.

4. Line 169 I recommend moving these definitions to earlier in the methods section. Perhaps you could rename the section entitled “Definition of a responder” to “Definitions” and then have sub-headings that distinguish between the different components of ‘ethical loss.’ Moreover, I found myself having to go back to the definitions several times as I was reading and encourage the authors to put together a definitions table for quick reference (if space permits).

5. Line 220-221 I would argue that whether a trial is ethical or not is entirely dependent on the trial’s risk-benefit analysis, not whether there is a positive benefit-loss for each participant (as long as risks to those participants are minimized). What’s more, the authors focus entirely on the direct benefits that a research participant might expect when participating in a trial (namely, getting access to a drug that works), however there are indirect benefits that are equally important, including participation in the production of generalizable knowledge. If the thrust of this paper is about the ethical impact of including ‘unnecessary participants’ in RCTs, I think it is critical to be clear about all of the relevant ethical considerations.

6. Line 212 In general, the manuscript tends to blur trial-level and participant-level risks and benefits, which is most apparent in the section entitled “Ethical issues and unnecessary participants.” Going back to my last point, if this is an ethics paper at heart, I encourage being very careful about the ethical considerations.

7. Line 331 The discussion section need significant work to highlight the meaning, importance and relevance of the results. In particular, the implications of the modeling need to be emphasized - why do the results matter?

The limitations section should follow the discussion.

7. PLOS authors have the option to publish the peer review history of their article (what does this mean?). If published, this will include your full peer review and any attached files.

Reviewer #2: **Yes: **Sylvester C. Chima, MD, LL.M, LLD

Reviewer #3: No

---

## [Editor Report · Acceptance letter]

27 Sep 2021

PONE-D-21-01394R1 

Using numerical modeling and simulation to assess the ethical burden in clinical trials and how it relates to the proportion of responders in a trial sample 

Dear Dr. Boissel:

I'm pleased to inform you that your manuscript has been deemed suitable for publication in PLOS ONE. Congratulations! Your manuscript is now with our production department. 

Kind regards, 

on behalf of

Dr. Dylan A Mordaunt 

Academic Editor

PLOS ONE